# Antibiotic Susceptibility of Bacterial Pathogens Stratified by Age in a Public Hospital in Qassim

**DOI:** 10.3390/healthcare10091757

**Published:** 2022-09-13

**Authors:** Saleh A. Alrebish, Nehad J. Ahmed, Hamed Al Hamed, Ajay Kumar, Hasan S. Yusufoglu, Amer Hayat Khan

**Affiliations:** 1Medical Education Department, College of Medicine, Qassim University, Buraydah 52571, Qassim, Saudi Arabia; 2College of Dentistry and Pharmacy, Buraydah Private Colleges, Buraydah 51418, Qassim, Saudi Arabia; 3Department of Clinical Pharmacy, Pharmacy College, Prince Sattam Bin Abdulaziz University, Al-Kharj 11942, Riyadh, Saudi Arabia; 4Discipline of Clinical Pharmacy, School of Pharmaceutical Sciences, Universiti Sains Malaysia, Gelugor 11800, Penang, Malaysia; 5Department of Laboratory Services, King Fahad Specialist Hospital, Buraydah 52366, Qassim, Saudi Arabia

**Keywords:** age, antibiogram, bacteria, stratification, susceptibility

## Abstract

Antibiotics have completely transformed medical practice by enabling the treatment of infections that were formerly fatal. However, misuse of antibiotics encourages the formation and spread of germs that are resistant to therapy, hastening the emergence of bacterial resistance. This was a retrospective study that aimed to gather information about the variation in bacterial susceptibility of various patient age groups in a public hospital in Qassim, Saudi Arabia from January 2020 to December 2021. The study included reviewing bacterial susceptibility results that were collected from the laboratory department of the hospital. Four thousand seven hundred and sixty-two isolates were collected. The age of 46.41% of the patients was more than 63 years and the age of 28.96% of the patients was less than 48 years. The most prevalent bacteria were Staphylococcus aureus, Escherichia coli, and Klebsiella pneumoniae. The resistance of gram-positive and gram-negative bacteria to different antibiotics in the elderly group was generally higher than the resistance rates in younger patients. For example, in patients less than 48 years old, the resistance of Staphylococcus haemolyticus to clindamycin (53.3%), ampicillin (91.4%), ciprofloxacin (68.2%), erythromycin (86.1%), and penicillin (93.18%) was high. In patients aged more than 63 years, Staphylococcus haemolyticus was highly resistant to sulfamethoxazole (54.8%), clindamycin (63.9%), ampicillin (98.1%), ciprofloxacin (79.1%), erythromycin (93.2%), gentamicin (63.6%), and penicillin (98.7%). Before prescribing the antibiotics, it is important to assess the microbes that patients have and to be aware of the bacterial isolates’ patterns of antibiotic susceptibility among patients of various age groups.

## 1. Introduction

Antibiotics have completely transformed medical practice by enabling the development of cancer chemotherapy, organ transplantation, and the treatment of diseases that were formerly fatal [1]. Early use of antibiotics in the course of sickness treatment can lower morbidity and even save lives. However, misuse of antibiotics encourages the formation and spread of germs that are resistant to therapy, hastening the emergence of bacterial resistance [1]. Multidrug resistance, extended resistance, and pandrug resistance are some of the different types of resistance that have developed over time [2]. Being resistant to all agents in all antimicrobial categories is referred to as pandrug resistance. “Multidrug resistance” is defined as acquired non-susceptibility to at least one agent in three or more antimicrobial categories. Non-susceptibility to at least one agent in all but two or fewer antimicrobial groups is referred to as extended resistance [3].

The World Health Organization classified antibiotic resistance as one of the top 10 dangers to global health in 2019 [4]. In the US, it is estimated that about 2.8 million antibiotic-resistant diseases occur each year, resulting in over 35,000 fatalities [2]. The misuse and incorrect use of antibiotics contribute to antimicrobial resistance, excessive healthcare expenditures, avoidable adverse drug events, and *Clostridioides difficile* infection [2].

To ensure that patients receive the appropriate antibiotic, an antibiogram might be employed. The antibiogram is a quick and convenient tool for determining the diseases and susceptibilities that are common in a particular institution. Antimicrobial stewardship practices are necessary, along with the traditional antibiogram, to guarantee that patients receive the appropriate antibiotic therapy based on a suspected site of infection, hospital location, and patient characteristics [5,6]. Together, clinical microbiologists and antimicrobial stewardship initiatives can create more intricate antibiograms that enhance empiric antibiotic treatment [5,6].

Cumulative antibiograms at a hospital can conceal differences in patient demographics, hospital departments, or anatomical areas. The information provided by hospital-wide cumulative antibiograms may not be sufficient to enable informed choices regarding the optimum care for hospitalized patients [5,6,7]. A syndromic antibiogram can be improved by grouping susceptibilities based on the patient’s location. The source of the sickness, how the infection was acquired, the location of the hospital, and patient characteristics like age can also be used to stratify the data [5,6].

Garcia et al. reported that the mode of action of the antibiotic affected the pattern of antibiotic resistance in relation to patient age. While antibiotics that target ribosomal functions (such as aminoglycosides) or cell wall synthesis (such as cephalosporin) do not exhibit an age-dependent pattern and have a consistent level of resistance across all age classes, antibiotics that target DNA synthesis (such as fluoroquinolones) exhibit a direct correlation with the age of patients, with higher rates of resistance among the older population [8]. Ji et al. stated that despite having a lower prevalence of Helicobacter pylori (H. pylori) infection, patients between the ages of 31 and 50 and the ages of 71 and 80 had greater rates of levofloxacin and clarithromycin resistance. Patients aged 71 to 80 showed the highest risk of antibiotic resistance. H. pylori resistance to clarithromycin and levofloxacin is significantly influenced by age. Therefore, personalized medication must be taken into account for the best care of individuals with H. pylori infection [9].

Globally, there is a lack of research on the variation in bacterial resistance across patients of different ages. Therefore, the objective of the current study was to examine the differences in bacterial susceptibility of different patient age groups in a public hospital in Qassim.

## 2. Materials and Methods

This was a retrospective study that was conducted to gather information about the variation in bacterial susceptibility of various patient age groups in a public hospital in Qassim, Saudi Arabia from January 2020 to December 2021. The study included reviewing bacterial susceptibility results that were collected from the laboratory department of the hospital.

The study includes the microbiology culture findings and sensitivity data of the microorganisms responsible for specific infections from patients admitted in 2020 and 2021. The study did not include the patients’ duplicate isolates. The information gathered comprised the total number of bacteria, the most common gram-negative and gram-positive bacteria in various patient age groups, and the rates of susceptibility for both gram-negative and gram-positive bacteria. An Excel sheet was used to collect and evaluate the data. Numbers and percentages were used to represent the results. Subtracting 100 percent from the susceptibility rate yields the resistance rate.

In the present study, we used the age group classification of the Lin et al. study. Age was classified into the following age groups: 0–14 years old (pediatric group), 15–47 years old (youth group), 48–63 years old (middle-aged group), and ≥64 years old (elderly group) [10]. However, because the number of patients in the pediatric group was low, we combined it with the youth group.

The data were collected after obtaining approval from the Regional Research Ethics Committee of the Qassim region with an approval number of 1443-862269.

## 3. Results

Four thousand seven hundred and sixty-two isolates were collected in the hospital during 2020 and 2021. The age of 28.96% of the patients was less than 48 years. The most prevalent bacteria in patients less than 48 years old were *Staphylococcus aureus* (19.36%), *Escherichia coli* (13.56%), *Klebsiella pneumoniae* (10.59%), and *Acinetobacter baumannii* (8.05%). The number and percentages of bacterial isolates in patients less than 48 years old are shown in Table 1.

The age of 24.63% of the patients was between 48 and 63 years. The most prevalent bacteria in patients aged between 48 and 63 years were *Staphylococcus aureus* (19.18%), *Klebsiella pneumoniae* (11.51%), *Escherichia coli* (11.08%), *Acinetobacter baumannii* (8.27%), and *Staphylococcus epidermidis* (6.22%) (Table 2).

The age of 46.41% of the patients was more than 63 years. The most prevalent bacteria in patients aged more than 63 years were *Klebsiella pneumoniae* (13.76%), *Staphylococcus aureus* (11.13%), *Escherichia coli* (10.63%), *Acinetobacter baumannii* (9.19%), and *Staphylococcus epidermidis* (8.01%) (Table 3).

Regarding gram-negative bacterial susceptibility in patients aged less than 48 years, *Pseudomonas aeruginosa* was highly resistant only to sulfamethoxazole (Resistance rate = 51.7%). *Proteus mirabilis* was highly resistant to ampicillin (76.5%) and to aztreonam (60.0%). The resistance of *Klebsiella pneumoniae* to ampicillin, aztreonam, ceftazidime, cefotaxime, and cefuroxime was more than 50%. The resistance of *Escherichia coli* to ampicillin (64.7%) was high. The resistance of *Acinetobacter baumannii* to all of the tested antibiotics (Table 4) was high.

In patients aged 48–63 years, *Pseudomonas aeruginosa* was highly resistant only to sulfamethoxazole (66.7%). The resistance of *Proteus mirabilis* to sulfamethoxazole (58.8%), ampicillin (82.61%), aztreonam (64.0%), ciprofloxacin (56.0%), and cefuroxime (51.1%) was high. The resistance of *Klebsiella pneumoniae* to sulfamethoxazole, ampicillin, aztreonam, ceftazidime, cefotaxime, and cefuroxime was more than 50%. The resistance of *Escherichia coli* to ampicillin (75.4%) and aztreonam (55.6%) was high. The resistance of *Acinetobacter baumannii* to all of the tested antibiotics was high. The resistance rates of *Enterobacter cloacae* to ampicillin (95.2%), aztreonam (55.6%), ceftazidime (51.2%), cefotaxime (56.4%), and cefuroxime (67.6%) were high (Table 5).

In patients aged more than 63 years, *Pseudomonas aeruginosa* was highly resistant only to sulfamethoxazole (63.6%). The resistance of *Proteus mirabilis* to all of the tested antibiotics except gentamicin was more than 50%. The resistance of *Escherichia coli* to ampicillin (76.8%), aztreonam (60.6%), ciprofloxacin (53.5%), and cefuroxime (50.7%) was high. The resistance rates of *Acinetobacter baumannii* and *Klebsiella pneumoniae* to all of the tested antibiotics were high. The resistance of *Enterobacter cloacae* to ampicillin (100.0%), aztreonam (63.6%), ceftazidime (57.7%), cefotaxime (68.6%), and cefuroxime (78.4%) was high (Table 6).

The results showed that, in general, the resistance rates of gram-negative bacteria to different antibiotics in the middle age group (48 to 63 years old) and the elderly group (more than 63 years old) were higher than the resistance rates in younger patients (less than 48 years old).

Regarding gram-positive bacterial susceptibility in patients less than 48 years old, the resistance of *Staphylococcus aureus* to ampicillin (77.6%) and penicillin (90.5%) was high. The resistance of *Staphylococcus haemolyticus* to clindamycin (53.3%), ampicillin (91.4%), ciprofloxacin (68.2%), erythromycin (86.1%), and penicillin (93.18%) was high. The resistance of *Staphylococcus hominis* to ampicillin (90.0%), erythromycin (71.2%), and penicillin (93.2%) was high. *Staphylococcus epidermidis* was highly resistant to ampicillin (92.5%), erythromycin (68.6%), and penicillin (90.1%). The resistance rates of *Enterococcus* species to sulfamethoxazole (78.3%), clindamycin (50.0%), and erythromycin (57.1%) were high (Table 7).

Regarding gram-positive bacterial susceptibility in patients aged 48 to 63 years, the resistance of *Staphylococcus aureus* to ampicillin (71.7%) and penicillin (89.3%) was high. *Staphylococcus hominis* was highly resistant to ampicillin (90.5%), erythromycin (57.5%), and penicillin (92.1%). *Staphylococcus epidermidis* was highly resistant to ampicillin (97,7%), ciprofloxacin (59.2%), erythromycin (71.8%), and penicillin (97.0%). *Enterococcus* species were highly resistant to sulfamethoxazole (85.7%), ciprofloxacin (54.3%), and erythromycin (68.4%) (Table 8).

Regarding gram-positive bacterial susceptibility in patients aged more than 63 years, the resistance of *Staphylococcus aureus* to ampicillin (72.9%) and penicillin (91.3%) was high. *Staphylococcus hominis* was highly resistant to ampicillin (89.4%), ciprofloxacin (60.5%), erythromycin (78.1%), and penicillin (90.1%). *Staphylococcus haemolyticus* was highly resistant to sulfamethoxazole (54.8%), clindamycin (63.9%), ampicillin (98.1%), ciprofloxacin (79.1%), erythromycin (93.2%), gentamicin (63.6%), and penicillin (98.7%). The resistance rates of *Staphylococcus auricularis* to clindamycin (57.1%), ampicillin (93.9%), ciprofloxacin (61.1%), erythromycin (51.1%), and penicillin (96.0%) were high.

*Staphylococcus epidermidis* was highly resistant to clindamycin (55.4%), ampicillin (97.1%), ciprofloxacin (71.9%), erythromycin (72.8%), and penicillin (97.5%). The resistance of *Enterococcus* species to sulfamethoxazole (85.71%), ciprofloxacin (65.3%), erythromycin (78.1%), and gentamicin (56.5%) was high. *Staphylococcus capitis* was highly resistant to clindamycin (52.0%), ampicillin (89.7%), ciprofloxacin (65.3%), erythromycin (51.0%), and penicillin (95.5%) (Table 9). In general, the resistance of gram-positive bacteria to different antibiotics in the elderly group was higher than the resistance rates in younger patients.

## 4. Discussion

The present study showed that several gram-positive bacteria such as *Staphylococcus epidermidis* and *Staphylococcus haemolyticus* were resistant to several antibiotics, mainly in elderly patients, but fortunately the resistance rates to linezolid and daptomycin were very low. Similarly, previous studies found that despite the rising usage of linezolid and daptomycin, resistance to these drugs has been identified but is still relatively rare [11,12,13].

The present study also showed that most of the gram-negative bacteria were resistant to several antibiotics, and that *Acinetobacter baumannii* and *Klebsiella pneumoniae* were resistant to almost all of the tested antibiotics in elderly patients. Similarly, previous studies showed that *Acinetobacter baumannii* and *Klebsiella pneumoniae* have high resistance rates to antibiotics [14,15]. Nonetheless, the resistance rates of gram-negative bacteria to several effective antibiotics such as amikacin, imipenem, meropenem, and colistin were not tested.

In general, the resistance of gram-positive and gram-negative bacteria to different antibiotics in the elderly group was higher than the resistance rates in younger patients. According to the National Institute for Public Health and the Environment, antibiotic-resistant bacteria are a problem, especially for weak individuals like the elderly [16]. Hossain et al. indicated that amikacin may be less effective for treating urinary tract infections in older individuals [17]. Moreover, Denkinger et al. reported that older patients are one of the hospital’s main reservoirs for multidrug-resistant infections [18]. Garcia et al. reported that as people age, the prevalence of isolates resistant to antibiotics that target DNA synthesis increases. As a preventative step to lower the incidence of resistant infections in each susceptible population, they emphasize the significance of patient age in the selection of antibiotics [8].

The Center for Infectious Disease Research and Policy indicated that because they spend longer than younger people in hospitals and long-term care institutions and have more chronic diseases, elderly individuals are more likely to be exposed to multidrug-resistant organisms and are at a higher risk for infections [19]. The Canadian Pediatric Society suggested that regional laboratories should regularly create and distribute age-specific antibiograms for use by physicians [20]. Klinker et al. reported that the hospital’s location, the source of the infection, and patient factors like age might all be used to stratify the susceptibility data [5]. Kumar et al. stated that most antimicrobial drugs’ activity decreased with aging, necessitating a change in empirical antibiotic therapy based on the patient’s age [21]. Yvonne reported that antibiotic resistance is common and affects the elderly, and that age-related anatomical and physiological changes, as well as the use of a urine catheter, nasogastric and percutaneous feeding tubes, and intravenous catheters predispose elderly patients to bacterial colonization and infections [22].

The percentage of male patients in the present study was higher than that of female patients. About 52.81% of the patients aged more than 63 years, 57.29% of the patients aged less than 47 years, and 60.10% of the patients aged between 48 and 63 years were male. Several previous studies found that the susceptibility rates of bacteria to antibiotics in male patients are different than those in female patients [23,24,25,26]. Further studies are needed to assess the difference in bacterial resistance between males and females.

Knowing the sources of samples is also important in the preparation of a stratified antibiogram. For example, the resistance rates of bacteria in a blood sample could differ from the resistance rates of bacteria that were isolated from urine samples. Kuster et al. reported that isolates from different anatomical sites showed variation in antimicrobial susceptibility rates [7]. The main limitation of the present study was that there was no information on the origin of the bacterial strains.

Bacterial co-infection is relatively infrequent in hospitalized COVID-19 patients [27]. Nevertheless, since the beginning of the COVID-19 pandemic, there has been growing concern over a potential rise in antimicrobial resistance secondary to increased antibiotic prescription for COVID-19 patients [28]. The second limitation of the present study was that it is unknown if the patients only had bacterial infections or if they also had COVID-19, which could cause secondary bacterial infections.

## 5. Conclusions

The resistance of gram-positive and gram-negative bacteria to different antibiotics in the elderly group was generally higher than the resistance rates of younger patients. It is important to assess the microbes that patients have and to be aware of the bacterial isolates’ patterns of antibiotic susceptibility among patients of various age groups before prescribing the antibiotics.

## Figures and Tables

**Table 1 healthcare-10-01757-t001:** The most prevalent bacteria in patients aged less than 48 years.

Bacteria	Number	Percentage
*Staphylococcus aureus*	267	19.36%
*Escherichia coli*	187	13.56%
*Klebsiella pneumoniae*	146	10.59%
*Acinetobacter baumannii*	111	8.05%
*Staphylococcus epidermidis*	89	6.45%
*Pseudomonas aeruginosa*	87	6.31%
*Enterococcus* Species	83	6.02%
*Staphylococcus hominis*	54	3.92%
*Staphylococcus haemolyticus*	46	3.33%
*Proteus mirabilis*	37	2.68%
Other bacteria *	272	19.72%
Total	1379	100.00%

* Other bacteria means the sum of the different bacteria that were collected infrequently (in less than 30 patients).

**Table 2 healthcare-10-01757-t002:** The most prevalent bacteria in patients aged between 48 and 63 years.

Bacteria	Number	Percentage
*Staphylococcus aureus*	225	19.18%
*Klebsiella pneumoniae*	135	11.51%
*Escherichia coli*	130	11.08%
*Acinetobacter baumannii*	97	8.27%
*Staphylococcus epidermidis*	73	6.22%
*Enterococcus* species	72	6.14%
*Pseudomonas aeruginosa*	67	5.71%
*Proteus mirabilis*	51	4.35%
*Staphylococcus hominis*	44	3.75%
*Enterobacter cloacae*	42	3.58%
Other bacteria *	237	20.20%
Total	1173	100.00%

* Other bacteria means the sum of the different bacteria that were collected infrequently (in less than 30 patients).

**Table 3 healthcare-10-01757-t003:** The most prevalent bacteria in patients aged more than 63 years.

Bacteria	Number	Percentage
*Klebsiella pneumoniae*	304	13.76%
*Staphylococcus aureus*	246	11.13%
*Escherichia coli*	235	10.63%
*Acinetobacter baumannii*	203	9.19%
*Staphylococcus epidermidis*	177	8.01%
*Enterococcus* species	157	7.10%
*Pseudomonas aeruginosa*	157	7.10%
*Staphylococcus hominis*	120	5.43%
*Proteus mirabilis*	93	4.21%
*Staphylococcus haemolyticus*	75	3.39%
*Staphylococcus auricularis*	55	2.49%
*Enterobacter cloacae*	54	2.44%
*Staphylococus capitis*	52	2.35%
Other bacteria *	282	12.76%
Total	2210	100.00%

* Other bacteria means the sum of the different bacteria that were collected infrequently (in less than 30 patients).

**Table 4 healthcare-10-01757-t004:** Gram-negative bacterial resistance rates in patients aged less than 48 years.

Bacterial Isolates	Sulfamethoxazole	Ampicillin	Aztreonam	Ciprofloxacin	Gentamicin	Ceftazidime	Cefotaxime	Cefuroxime
*Pseudomonas aeruginosa*	51.7	NA	35.7	23.5	11.5	19.0	NA	20.0
*Proteus mirabilis*	45.9	76.5	60.0	40.0	28.6	36.1	36.1	41.2
*Klebsiella pneumoniae*	39.7	97.3	66.1	48.9	35.0	54.9	55.2	56.7
*Escherichia coli*	41.4	64.7	40.6	28.5	9.4	29.4	30.3	29.9
*Acinetobacter baumannii*	66.7	NA	100.0	88.0	56.3	83.2	100.0	100.0

**Table 5 healthcare-10-01757-t005:** Gram-negative bacterial resistance rates in patients aged 48–63 years.

Bacterial Isolates	Sulfamethoxazole	Ampicillin	Aztreonam	Ciprofloxacin	Gentamicin	Ceftazidime	Cefotaxime	Cefuroxime
*Pseudomonas aeruginosa*	66.7	NA	32.4	25.0	15.5	25.4	NA	NA
*Proteus mirabilis*	58.8	82.6	64.0	56.0	42.0	44.9	47.9	51.1
*Klebsiella pneumoniae*	54.1	97.3	68.2	48.8	35.9	56.0	56.9	60.3
*Escherichia coli*	47.3	75.4	55.6	42.5	18.7	40.8	40.3	45.2
*Acinetobacter baumannii*	57.3	100.0	100.0	90.7	57.8	90.1	90.9	100.0
*Enterobacter cloacae*	21.4	95.2	55.6	41.0	21.9	51.2	56.4	67.6

**Table 6 healthcare-10-01757-t006:** Gram-negative bacterial resistance rates in patients aged more than 63 years.

Bacterial Isolates	Sulfamethoxazole	Ampicillin	Aztreonam	Ciprofloxacin	Gentamicin	Ceftazidime	Cefotaxime	Cefuroxime
*Pseudomonas aeruginosa*	63.6	NA	36.2	24.5	19.6	25.2	NA	NA
*Proteus mirabilis*	64.1	76.0	67.5	65.6	46.3	51.6	54.6	59.1
*Klebsiella pneumoniae*	56.8	96.4	76.2	60.6	50.8	65.2	67.3	68.5
*Escherichia coli*	50.0	76.8	60.6	53.5	19.8	46.1	48.0	50.7
*Acinetobacter baumannii*	67.2	100.0	94.7	94.9	67.2	90.4	96.7	100.0
*Enterobacter cloacae*	33.3	100.0	63.6	40.4	21.6	57.7	68.6	78.4

**Table 7 healthcare-10-01757-t007:** Gram-positive bacterial resistance rates in patients less than 48 years old.

Bacterial Isolates	Sulfamethoxazole	Mupirocin	Linezolid	Clindamycin	Ampicillin	Ciprofloxacin	Erythromycin	Gentamicin	Penicillin	Daptomycin
*Staphylococcus aureus*	7.1	8.0	0.8	15.1	77.6	30.9	22.9	13.6	90.5	4.3
*Staphylococcus haemolyticus*	35.6	13.0	0.0	53.3	91.4	68.2	86.0	42.5	93.2	0.0
*Staphylococcus hominis*	29.6	26.9	0.0	30.2	90.0	42.6	71.1	14.3	93.2	NA
*Staphylococcus epidermidis*	19.3	30.6	2.3	40.2	92.4	42.4	68.6	23.7	90.1	0.0
*Enterococcus Species*	78.3	NA	1.3	50.0	35.7	25.0	57.1	21.9	39.2	0.0

**Table 8 healthcare-10-01757-t008:** Gram-positive bacterial resistance rates in patients aged 48–63 years.

Bacterial Isolates	Sulfamethoxazole	Mupirocin	Linezolid	Clindamycin	Ampicillin	Ciprofloxacin	Erythromycin	Gentamicin	Penicillin	Daptomycin
*Staphylococcus aureus*	6.2	8.9	1.8	17.5	71.7	34.8	23.4	10.2	89.3	0.0
*Staphylococcus hominis*	23.3	20.0	2.4	23.1	90.5	40.5	57.5	5.1	92.1	0.0
*Staphylococcus epidermidis*	32.9	16.7	1.4	42.9	97.7	59.1	71.8	39.3	97.0	0.0
*Enterococcus* Species	85.7	NA	3.1	NA	42.4	54.3	68.4	40.9	49.2	0.0

**Table 9 healthcare-10-01757-t009:** Gram-positive bacterial susceptibility in patients aged more than 63 years.

Bacterial Isolates	Sulfamethoxazole	Mupirocin	Linezolid	Clindamycin	Ampicillin	Ciprofloxacin	Erythromycin	Gentamicin	Penicillin	Daptomycin
*Staphylococcus aureus*	11.5	5.9	1.6	20.2	72.9	34.9	25.6	26.5	91.2	0.0
*Staphylococcus hominis*	30.0	17.0	0.8	48.3	89.4	60.5	78.1	12.4	90.1	0.0
*Staphylococcus haemolyticus*	54.8	11.1	2.7	63.9	98.1	79.1	93.2	63.6	98.6	0.0
*Staphylococcus auricularis*	23.6	30.4	7.7	57.1	93.9	61.1	51.1	39.6	96.0	0.0
*Staphylococcus epidermidis*	39.4	38.4	4.6	55.4	97.1	71.9	72.8	45.1	97.5	0.0
*Enterococcus* species	85.7	NA	2.7	44.4	39.1	65.3	78.1	56.5	39.5	1.1
*Staphylococcus capitis*	9.6	17.6	8.0	52.0	89.7	65.3	51.0	47.8	95.4	0.0

## Data Availability

Not applicable.

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
