# Peer review of "Antibiotic Susceptibility of Bacterial Pathogens Stratified by Age in a Public Hospital in Qassim"

_healthcare, 2022, doi:10.3390/healthcare10091757_

Round 1
Reviewer 1 Report
It is an interesting retrospective study on the spread of antibiotic resistance among different age groups in a public hospital in Saudi Arabia from January 2020 to December 2021.
However, the study should address the following points:
Although data was collected between January 2020 and December 2021 during COVID-19 pandemic authors have not referred to this at all – more data should be included if any of the isolates collected from COVID positive patients due to secondary bacterial infections
·
· Which samples were collected from patients? – Authors need to mention the source of bacterial strains (blood, sputum, etc.)
· Were bacterial isolates associated with or responsible for certain disease/s (septicaemia, urinary tract infection, etc.)? or they were just carried by patients?
· Does this study include any children/infants? – it is not clear if the group (less than 48 years old include children or not)- Authors should provide details of the spread of resistant strains in children.
Discussion section should discuss in depth the significance of results after addressing above points and including sources of samples and clarifying if any isolates collected from children……….
Other comments:
The abstract should include more detailed results with the percentage of resistance among certain age groups.
The introduction should refer to previous studies reported resistance in different age groups for example; https://pubmed.ncbi.nlm.nih.gov/29116037/ , https://pubmed.ncbi.nlm.nih.gov/26937912/
Bacterial names must be written in italics
Few grammar mistakes; for example, data was (line 81)
Results can be presented in a better way – some of the tables can be combined.
Author Response
It is an interesting retrospective study on the spread of antibiotic resistance among different age groups in a public hospital in Saudi Arabia from January 2020 to December 2021.
However, the study should address the following points:
Although data was collected between January 2020 and December 2021 during COVID-19 pandemic authors have not referred to this at all – more data should be included if any of the isolates collected from COVID positive patients due to secondary bacterial infections
I also mention the drawback that there was no information on the origin of the bacterial strains. Furthermore, it is unknown which COVID positive individuals' isolates were caused by subsequent bacterial infections.
- Which samples were collected from patients? – Authors need to mention the source of bacterial strains (blood, sputum, etc.)
I also mention the drawback that there was no information on the origin of the bacterial strains. Furthermore, it is unknown which COVID positive individuals' isolates were caused by subsequent bacterial infections.
- Were bacterial isolates associated with or responsible for certain disease/s (septicaemia, urinary tract infection, etc.)? or they were just carried by patients?
I add that the study included the bacterial isolates that were responsible for infections
- Does this study include any children/infants? – it is not clear if the group (less than 48 years old include children or not)- Authors should provide details of the spread of resistant strains in children.
I add a paragraph in the methodology that the number of pediatric patients was low, so we combined them with the youth. I add the detailed age classification in the methodology.
Discussion section should discuss in depth the significance of results after addressing above points and including sources of samples and clarifying if any isolates collected from children……….
I add two paragraph in the discussion about COVID 19 coinfections and source of samples and I wrote two limitations of the study. I wrote about the pediatrics and age classifications in the methodology section.
Other comments:
The abstract should include more detailed results with the percentage of resistance among certain age groups.
I add a sentence about the percentage of patients in each group and I add sentences as an example for the change in the resistance rates of Staphylococcus haemolyticus.
The introduction should refer to previous studies reported resistance in different age groups for example; https://pubmed.ncbi.nlm.nih.gov/29116037/ , https://pubmed.ncbi.nlm.nih.gov/26937912/
I add a paragraph in the introduction about previous studies that reported resistance in different age groups
Bacterial names must be written in italics
I write bacterial names in italic
Few grammar mistakes; for example, data was (line 81)
I modify it and check the manuscript again
Results can be presented in a better way – some of the tables can be combined.
If you want, I will combine but the ordering of the most common bacteria was different between age groups and regarding the resistance of bacteria the tested antibiotics were different in gram-positive bacteria from gram-negative bacteria.
Reviewer 2 Report
Your paper is interesting and well presented, but some modifications are necessary.
1. When you report the name of a microrganism, the genus is on Capital letter, the species in lowercase letter and in Italic; i.e. Pseudomonas aeruginosa, Staphylococcus epidermidis etc. Please, uniform the name of bacteria according this international rule.
2. In tables the susceptibility of different bacteria to antimicrobial drugs is reported, but in results you described the resistance. This way of results description is particularly confusing. Generally, in the results are described exactly what you read in the table, not the inverse number; tables should be a way to highlight and simplify the reading of the results. I think it will be better uniform data report in tables and data described in the Results, or all susceptibility data or all resistance data. In Discussion you can report and discuss between susceptibility and resistance.
3. Why you subdivided patients in these three groups? It would have been interesting if they had all been women and consider the difference between pre-menopause, menopause and postmenopause. By the way, how many are women and how many are men? It will be interesting reports the differences by sex...
Author Response
Your paper is interesting and well presented, but some modifications are necessary.
- When you report the name of a microrganism, the genus is on Capital letter, the species in lowercase letter and in Italic; i.e. Pseudomonas aeruginosa,Staphylococcus epidermidis Please, uniform the name of bacteria according this international rule.
I uniformed the name of bacteria according to the rule.
- In tables the susceptibility of different bacteria to antimicrobial drugs is reported, but in results you described the resistance. This way of results description is particularly confusing. Generally, in the results are described exactly what you read in the table, not the inverse number; tables should be a way to highlight and simplify the reading of the results. I think it will be better uniform data report in tables and data described in the Results, or all susceptibility data or all resistance data. In Discussion you can report and discuss between susceptibility and resistance.
I change the table percentages from susceptibility rates to resistance rate
- Why you subdivided patients in these three groups? It would have been interesting if they had all been women and consider the difference between pre-menopause, menopause and postmenopause. By the way, how many are women and how many are men? It will be interesting reports the differences by sex...
I add a paragraph about the number of females
I add also a paragraph in the methodology for the age classification that we used.